# Leveraging Demonstrations with Latent Space Priors

**Jonas Gehring**                                                    *jgehring@meta.com*
*Meta AI (FAIR), ETH Zürich*

**Deepak Gopinath**                                               *dgopinath@meta.com*
**Jungdam Won**                                                    *jungdam@meta.com*
*Meta AI (FAIR), Pittsburgh, PA*

**Andreas Krause**                                                  *krausea@ethz.ch*
*ETH Zürich*

**Gabriel Synnaeve**                                                   *gab@meta.com*
**Nicolas Usunier**                                                 *usunier@meta.com*
*Meta AI (FAIR), France*

**Reviewed on OpenReview:** *https://openreview.net/forum?id=OzGIu4T4Cz*

## Abstract

Demonstrations provide insight into relevant state or action space regions, bearing great potential to boost the efficiency and practicality of reinforcement learning agents. In this work, we propose to leverage demonstration datasets by combining skill learning and sequence modeling. Starting with a learned joint latent space, we separately train a generative model of demonstration sequences and an accompanying low-level policy. The sequence model forms a latent space prior over plausible demonstration behaviors to accelerate learning of high-level policies. We show how to acquire such priors from state-only motion capture demonstrations and explore several methods for integrating them into policy learning on transfer tasks. Our experimental results confirm that latent space priors provide significant gains in learning speed and final performance. We benchmark our approach on a set of challenging sparse-reward environments with a complex, simulated humanoid, and on offline RL benchmarks for navigation and object manipulation[1]

## 1 Introduction

Recent results have demonstrated that reinforcement learning (RL) agents are capable of achieving high performance in a variety of well-defined environments (Silver et al., 2016; Vinyals et al., 2019; OpenAI et al., 2019; Badia et al., 2020). A key requirement of these systems is the collection of billions of interactions via simulation, which presents a major hurdle for their application to real-world domains. Therefore, it is of high interest to develop methods that can harness pre-trained behaviors or demonstrations to speed up learning on new tasks via effective generalization and exploration.

In this work, we set out to leverage demonstration data by combining the acquisition of low-level motor skills and the construction of generative sequence models of state trajectories. In our framework, low-level skill policies reduce the complexity of downstream control problems via action space transformations, while state-level sequence models capture the temporal structure of the provided demonstrations to generate distributions over plausible future states. We ground both components in a single latent space so that sequence model outputs can be translated to environment actions by the low-level policy. Thus, our sequence models constitute useful priors when learning new high-level policies.

---

[1]Videos, source code and pre-trained models available at `https://facebookresearch.github.io/latent-space-priors`.

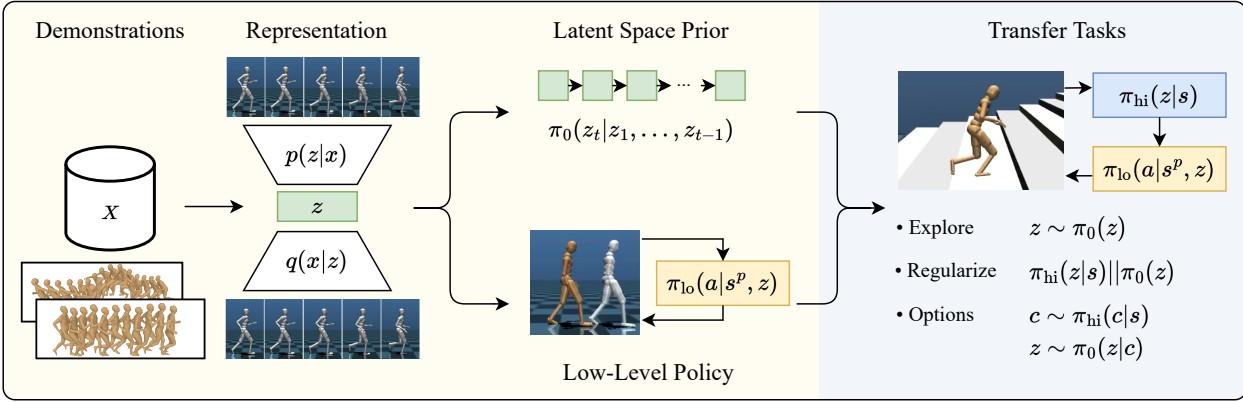

Figure 1: Our approach consists of a pre-training phase (left), followed by high-level policy learning on transfer tasks. For pre-training, we embed demonstration trajectories $\boldsymbol{x} \in X$ into a latent representations $z$ with an auto-encoder. We separately learn a prior $\pi_0$ that models latent space trajectories, as well as a low-level policy $\pi_{\text{lo}}$ trained to reenact demonstrations from proprioceptive observations $s^p$ a near-term targets $z$. On transfer tasks, we train a high-level policy $\pi_{\text{hi}}(z|s)$ and utilize the latent space prior $\pi_0$ to accelerate learning.

Our latent space priors empower RL agents to address several key challenges for efficient learning. Directed exploration helps to discover state regions with high rewards and is achieved with an augmented exploration policy that samples latent state sequences from the prior. Similarly, they offer additional learning signals by regularizing policies, with the prior distribution encoding behavior that is in correspondence with the provided demonstrations. Finally, latent state priors generate temporally extended behaviors, providing flexible temporal abstractions in the form of variable-length options, without relying on fixed temporal partitioning of the training data.

Figure 1 summarizes our approach, which we showcase on continuous control tasks with a complex humanoid robot. Our demonstrations are publicly available motion capture sequences that comprise observations of general locomotion skills. First, we learn a latent space of demonstration states by auto-encoding trajectories. Latent space priors and low-level policies are then acquired separately. The prior models demonstrations as sequences of latent states and is implemented as an auto-regressive Transformer (Vaswani et al., 2017). The low-level policy is trained to reenact the provided demonstration clips in a simulated environment (Hasenclever et al., 2020; Won et al., 2021), where latent states constitute short-term goals for conditioning. On transfer tasks, we train new high-level policies from scratch utilizing pre-trained components, i.e., latent state priors and low-level skill policies. We investigate several methods of integration: (1) augmenting the exploration policy with sequences sampled from the prior; (2) regularizing the policy towards distributions predicted by the prior; (3) conditionally generating sequences from high-level actions actions to provide temporal abstraction.

We evaluate our approach on a set of sparse-reward tasks that pose varying challenges with respect to exploration, long-term reasoning and skill transfer. We find that latent space priors provide benefits across all these axes and lead to improved learning speed and final performance. Depending on the concrete task at hand, the optimal way of utilizing latent space priors differs: temporal abstraction boosts exploration capabilities, while augmenting exploration and regularizing the high-level policy towards demonstration behavior accelerates learning and yields high final performance on tasks demanding quick reactions.

## 2 Latent Space Priors

### 2.1 Setting

We consider a reinforcement learning agent that seeks to maximize its cumulative reward by acting in a given task modelled as a Markov decision process $(\mathcal{S}, \mathcal{A}, T, r, \gamma)$(Sutton and Barto, 2018). Our approach is situated in a hierarchical reinforcement learning framework, where a high-level policy $\pi_{\text{hi}} : \mathcal{S} \to \mathcal{Z}$ steers a fixed, pre-trained low-level policy $\pi_{\text{lo}} : \mathcal{S} \times \mathcal{Z} \to \mathcal{A}$ via actions from a latent space $z \in \mathcal{Z}$ (Dayan and Hinton, 1992;

Vezhnevets et al., 2017). We now define a **latent space prior** as a generative, auto-regressive model in the high-level action space $\mathcal{Z}$, i.e., $\pi_0(z_t|z_1, \ldots, z_{t-1})$. Intuitively, a latent space prior models plausible high-level behavior, and we aim to show-case how it can be employed to improve high-level policy learning on new tasks.

A crucial insight is that both the low-level policy $\pi_{\mathrm{lo}}$ as well as the latent space prior $\pi_0$ can be acquired from demonstrations. Accordingly, our exposition targets a setting in which we are provided with a dataset $X$ consisting of observation trajectories $\boldsymbol{x} = (x_1, \ldots, x_n)$. The dataset is assumed to be unlabeled (i.e., not annotated with rewards) and unstructured (i.e., trajectories may include several tasks or skills, which may further be unrelated to our main tasks of interest). We structure our training setup such that latent states $z$ encode successive demonstration states. In effect, the latent space prior serves as a generative model of demonstration trajectories, while the low-level policy is conditioned on encoded near-term demonstration states.

We implement our method for continuous control of a simulated humanoid robot and leverage motion capture demonstrations. While we do not require an exact match between the parameterization of demonstrations and task states, we assume a correspondence between a demonstration state $x$ and proprioceptive states $s^p \in \mathcal{S}^{p\,2}$. In our experimental setup, proprioceptive states $s^p$ correspond to local sensor readings of a simulated humanoid robot, and transfer tasks of interest may provide additional state information, e.g., the global position of the robot, goal states or extra sensor readings. We provide a mapping of $\mathcal{X}$ to $\mathcal{S}^p$ which we describe in Appendix A. In the remainder of this section, we will detail the acquisition of a prior $\pi_0$ and a corresponding low-level-policy $\pi_{\mathrm{lo}}$ from demonstrations.

## 2.2 Modeling Demonstration Trajectories

We learn a generative model for demonstration trajectories in a two-step process, inspired by recent work in image and audio generation (van den Oord et al., 2017; Dhariwal et al., 2020). First, we learn a latent space $\mathcal{Z}$ of demonstration observations $x$ via auto-encoding. We then learn a separate, auto-regressive model (the latent space prior) of latent state sequences.

Demonstration trajectories are embedded with a variational auto-encoder (VAE) (Kingma and Welling, 2014), optimizing for both reconstruction accuracy and latent space disentanglement. Both encoder and decoder consist of one-dimensional convolutional layers such that each latent state covers a short window of neighboring states. Latent states thus capture short-range dynamics of the demonstration data. This eases the sequence modeling task, but is also beneficial to endow the low-level policy $\pi_{\mathrm{lo}}(a|s, z)$ with accurate near-term dynamics (2.3). The training loss of a demonstration sub-trajectory $\boldsymbol{x} = (x_{i-k}, \ldots, x_{i+k})$ of length $2k + 1$ is the evidence lower bound (ELBO),

$$\mathcal{L}_{\mathrm{AE}}(\boldsymbol{x}, \phi, \theta) = - \mathbb{E}_{q_\phi(\boldsymbol{z}|\boldsymbol{x})} \left[ \log p_\theta(\boldsymbol{x}|\boldsymbol{z}) \right] + \beta D_{\mathrm{KL}} \left( q_\phi(\boldsymbol{z}|\boldsymbol{x}) \,||\, p(\boldsymbol{z}) \right),$$

where $q_\phi$ is the encoder (parameterized by $\phi$), $p_\theta(\boldsymbol{x}|\boldsymbol{z})$ the decoder and $p(\boldsymbol{z})$ a prior, which we fix to an isotropic Gaussian distribution with unit variance.

VAEs are typically trained with a L2 reconstruction loss, corresponding to the ELBO when assuming $p_\theta$ to be Gaussian. Domain-specific reconstruction losses may however significantly enhance sample quality (Dhariwal et al., 2020; Li et al., 2021). For our application, we employ a forward kinematic layer (Villegas et al., 2018) to compute global joint locations from local joint angles in $x$. Global joint locations, position and orientation of reconstruction and reference are then scored with a custom loss function (Appendix C.1).

After training the VAE on the demonstration corpus, we map each trajectory $\boldsymbol{x} = (x_1, \ldots, x_n)$ to a latent space trajectory $\boldsymbol{z}$ with the encoder $q_\theta$. We then model trajectories in latent space with an auto-regressive prior, i.e.,

$$p(\boldsymbol{z}) = \prod_{t=1}^{n} \pi_0(z_t|z_1, \ldots, z_{t-1}) \ .$$

We implement latent space priors as Transformers (Vaswani et al., 2017), with the output layer predicting a mixture distribution of Gaussians (Bishop, 1994; Graves, 2013). As an alternative to employing dropout

---

[2]Our low-level policy operates on proprioceptive states $\mathcal{S}^p \subseteq S$ (Marino et al., 2019).

in embedding layers for regularization, we found that the variability and quality of the generated trajectories, transformed into poses with the VAE decoder, is significantly increased by sampling past actions from $q_\phi(\boldsymbol{z}|\boldsymbol{x})$ while predicting the mean of the latent distribution via teacher forcing. Training is further stabilized with an entropy term. The full loss for training the prior with neural network weights $\psi$ is

$$\mathcal{L}_{\text{Prior}}(\boldsymbol{x}, \psi) = - \mathop{\mathbb{E}}_{q_\phi(\boldsymbol{z}|\boldsymbol{x})} \left[ \sum_{t=1}^{n} \mathop{\mathbb{E}}_{z_t} \left[ \log \pi_{0,\psi}(z_t|z_1, \ldots, z_{t-1}) \right] + \epsilon H(\pi_{0,\psi}(\cdot|z_1, \ldots, z_{t-1})) \right] .$$

### 2.3 Low-Level Policy Learning

We train low-level skill policies to re-enact trajectories from the demonstrations, presented as latent state sequences $\{\boldsymbol{z}\}$. If the dataset includes actions, behavior cloning or offline RL techniques may be employed to reduce computational requirements (Lange et al., 2012; Merel et al., 2019b; Levine et al., 2020; Wagener et al., 2022). We apply our method to such a setting in Section 4.3. In our main experiments, demonstrations consist of observations only; we therefore employ example-guided learning, relying on an informative reward function for scoring encountered states against their respective demonstration counterparts (Peng et al., 2018; Won et al., 2020; Hasenclever et al., 2020).

Our training setup is closely related to the pre-training procedure in CoMic Hasenclever et al. (2020). A semantically rich per-step reward measures the similarity between the current state $s_t$ and the respective reference state $x_t$, which we ground in the simulation environment as detailed in Appendix A. Previous works on skill learning from motion capture data condition the policy on several next demonstration frames, featurized relative to the current state (Hasenclever et al., 2020; Won et al., 2020). In contrast, our skill policy is conditioned on a single latent state, which models near-term dynamics around a given time step in the demonstration trajectory (2.2). Recalling that each VAE latent state is computed from a window of $2k+1$ neighboring states, we provide the latent state $z_{t+k}$ to the skill policy at time step $t$. We use V-MPO (Song et al., 2019), an on-policy RL algorithm, to learn our low-level policy.

## 3 Accelerated Learning on Transfer Tasks

On transfer tasks, our main objective is to learn a high-level policy $\pi_{\text{hi}}$ efficiently, i.e., without requiring exceedingly many environment interactions and achieving high final returns. We employ Soft Actor-Critic (SAC), a sample-efficient off-policy RL algorithm (Haarnoja et al., 2018). While our pre-trained low-level policy is capable of re-enacting trajectories from the demonstration data, the high-level policy still has to explore the latent space in order to determine suitable action sequences for a given task. Latent space priors assist high-level policy learning by providing flexible access to distributions over trajectories in the policy's action space $\mathcal{Z}$. We will now detail several means to integrate a prior $\pi_0(z_t|z_1, \ldots z_{t-1})$ into the training process of $\pi_{\text{hi}}(z_t|s_t)$ (Figure 2), which we evaluate on several sparse-reward transfer tasks in Section 4.

### 3.1 Exploration

As our prior generates latent state sequences that, together with the the low-level policy, result in behavior corresponding to the provided demonstration, it can be directly utilized as an exploration policy. This augments standard exploration schemes in continuous control, where actions are obtained via additive random noise or by sampling from a probabilistic policy. While acting according to $\pi_0$ exhibits temporal consistency, the resulting behavior is independent of the specific RL task at hand as it does not take ground-truth simulation states or rewards into account. Consequently, $\pi_0$ is most useful in the early stages of training to increasing coverage of the state space. We thus augment exploration in an $\epsilon$-greedy manner and anneal $\epsilon$ towards 0 during training.

We further correlate exploration by following the actions of either $\pi_{\text{hi}}$ or $\pi_0$ for several steps. A low $\epsilon$ naturally results in extended periods of acting with the high-level policy; when exploring with the latent space prior, we sample the number of steps $k$ to follow the generated sequence from a Poisson distribution, where $k$'s

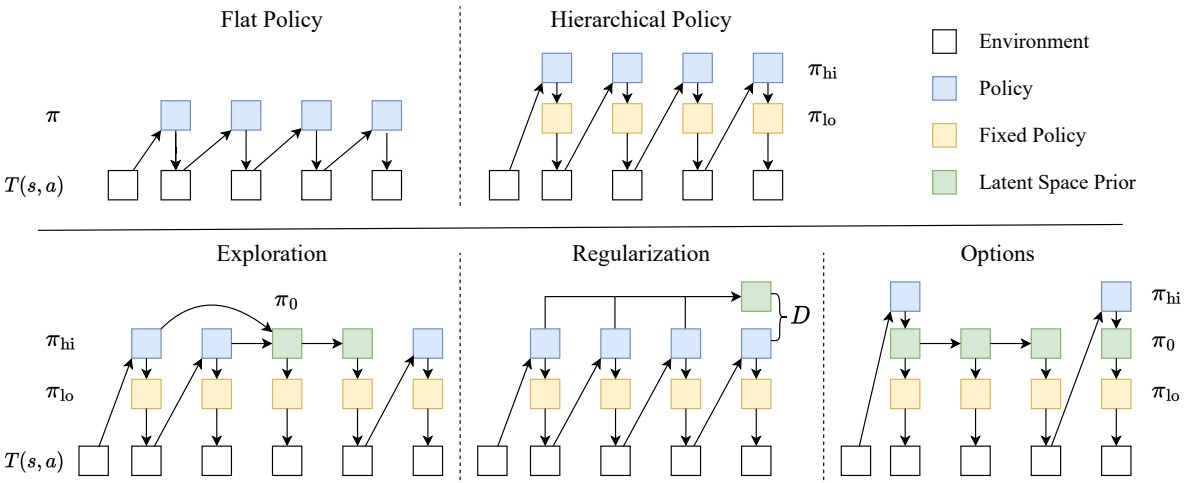

Figure 2: Top: graphical models of learning a policy (blue) without and with a pre-trained low-level policy (yellow). For clarity, we omit the low-level policy's dependency on the environment state. Bottom: comparison of methods to accelerate learning a high-level policy $\pi_{\text{hi}}$ (blue) with a latent state prior $\pi_0$ (green): augmenting exploration with actions from the prior (left); regularizing $\pi_{\text{hi}}$ towards behavior encapsulated by the prior (center); performing temporally extended actions with $\pi_0$ (right).

expectation is defined by a hyper-parameter $\lambda$. Formally, the augmented exploration policy is defined as

$$\pi_{\text{hi}}^{\text{exp}}(z_t|s_t) = \begin{cases} \pi_{\text{hi}}(z_t|s_t) & \text{with probability } 1 - \epsilon \\ \pi_0(z_t|z_{i<t}) & \text{with probability } \epsilon \quad \text{for } k \sim \text{Pois}(\lambda) \text{ steps} \end{cases}$$

### 3.2 Regularization

Reinforcement learning algorithms commonly employ inductive biases in the form of regularization terms in policy or value function optimization, with entropy maximization as a popular example (Ziebart et al., 2008; Ahmed et al., 2019). If prior knowledge is available in the form of a policy, it can form an additional bias, with a modified high-level policy learning objective given as

$$\max_{\pi_{\text{hi}}} \sum_t \mathbb{E}_{(s_t,a_t)\sim\pi_{\text{hi}}} \left[ r(s_t, a_t) - \delta D\left(\pi_{\text{hi}}(\cdot|s_t), \pi_0(\cdot|z_{i<t})\right) \right] \ .$$

Here, $a_t$ is provided by the deterministic low-level policy $\mathbb{E}_a[\pi_{\text{lo}}(a|s, z)]$, $D$ is a suitable divergence function (e.g., the KL divergence), and the scalar $\delta$ trades off reward maximization and prior regularization (Tirumala et al., 2020; Pertsch et al., 2020). We extend the SAC loss for policy optimization (Haarnoja et al., 2018, Eq. 7) as follows for network parameters $\rho$, with $B$ denoting the replay buffer and $\alpha$ the temperature coefficient for entropy regularization:

$$J_{\pi_{\text{hi}}}(\rho) = \mathbb{E}_{s_t\sim B} \left[ \mathbb{E}_{\substack{z_t\sim\pi_{\text{hi},\rho}(\cdot|s_t), \\ z_t^0\sim\pi_0(\cdot|z_{i<t})}} \left[ \alpha \log \pi_{\text{hi},\rho}(z_t|s_t) + \delta \log \pi_{\text{hi},\rho}(z_t^0|s_t) - Q_{\text{hi}}(s_t, a_t) \right] \right] \ .$$

Our formulation follows from several considerations. As $\pi_0(z_t|z_{i<t})$ is a mixture distribution over plausible future latent states, we aim for matching its support rather than its mode and hence opt for the forward KL divergence. We further leave out the normalization term $\log \pi_0(z_t^0|z_{i<t})$, which we found to not yield significant benefits. In line with Tirumala et al. (2020) but in contrast to Pertsch et al. (2020), we retain SAC's entropy regularization objective as we found that it provides significant benefits for exploration.

Similar to using latent space priors as an exploration policy in 3.1, their independence of task-specific rewards renders them most useful in the early stages of training when significant task reward signals are lacking.

Hence, we likewise anneal $\delta$ towards 0 as training progresses. Depending on the concrete setting, it may however be worthwhile to restrict behavior to demonstrations over the whole course of training, e.g., to avoid potentially destructive actions or to obtain behavior that matches the demonstration dataset closely.

### 3.3 Options

Hierarchical decomposition of search problems is a powerful concept in planning algorithms (Sacerdoti, 1974) and has found numerous applications in reinforcement learning (Dayan and Hinton, 1992; Sutton et al., 1999). The utility of such decomposition rests on problem-specific abstractions over some or all of the agent's state and action space, and on the temporal horizon of decision-making. In our framework, the joint latent space of $\pi_0$ and $\pi_{\text{lo}}$ constitutes an action space abstraction. Latent space priors further offer temporally extended actions, or options (Sutton et al., 1999), via conditional auto-regressive generation of latent state sequences. Consequently, the high-level policy now provides the conditional input to the prior (Fig. 2, right). We investigate a basic form of conditioning by predicting an initial latent state $z_t$ from which the the prior then predicts the following $z_{t+1}, \ldots, z_{t+k-1}$ high-level actions for an option length of $k$ steps.

For learning $\pi_{\text{hi}}$ with temporally extended actions, we modify the SAC objectives with the step-conditioned critic proposed by Whitney et al. (2020). This yields the following objectives, with $0 \le i < k$ denoting the number of environment steps from the previous high-level action:

$$J_{Q_{\text{hi}}}(\rho) = \mathbb{E}_{(i,s_t,a_t,z_{t-i})\sim B} \left[ \frac{1}{2} \left( Q_{\text{hi},\rho}(s_t, z_{t-i}, i) - \sum_{j=0}^{k-i-1} \left( \gamma^j r(s_{t+j}, a_{t+j}) \right) + \gamma^{k-i} V(s_{t+k-i}) \right)^2 \right]$$

$$J_{\pi_{\text{hi}}}(\rho) = \mathbb{E}_{s_t \sim B} \left[ \mathbb{E}_{z_t \sim \pi_{\text{hi},\rho}} \left[ \alpha \log \pi_{\text{hi},\rho}(z_t|s_t) - Q_{\text{hi}}(s_t, z_t, 0) \right] \right] .$$

### 3.4 Planning

If demonstrations are closely related to the task of interest, latent space priors may be used for planning in order to act without learning a high-level policy. Concretely, the learned auto-encoder can provide a latent representation of the trajectory to initialize new high-level action sequences. Sampled candidate sequences from $\pi_0$ can be decoded again into states, which are then ranked by a given reward function. We demonstrate successful planning for a simple navigation task in Appendix F. However, the main focus of our study are transfer settings where task dynamics may differ from demonstrations, e.g., due to additional objects, and demonstrations are obtained from real-world data.

We note that the described methods for integrating latent space priors into downstream policy learning are complementary so that, e.g., $\pi_0$ can assist in exploration while also providing a regularization objective, or sampled latent space sequences can be filtered based on their utility if the reward function is known. In our experiments, we evaluate each strategy in isolation to highlight its effects on learning performance.

## 4 Experiments

We evaluate the efficacy of our proposed latent space priors on a set of transfer tasks, tailored to a simulated humanoid robot offering a 56-dimensional action space (Hasenclever et al., 2020). Our demonstration dataset contains 20 minutes of motion capture data of various locomotion behaviors such as walking, turning, running and jumping (Appendix A). We consider four sparse-reward tasks that pose individual challenges with respect to locomotion, exploration and planning abilities (Fig. 3; additional details in Appendices B and G). We investigate the effect of latent space priors by comparing against a baseline which also uses our low-level policy but learns a high-level policy without extra support. We further compare against a low-level policy acquired with CoMic (Hasenclever et al., 2020), in which the high-level action space is learned jointly with the low-level policy.

The utilization of latent space priors for high-level policy learning introduces several hyper-parameters. For exploration (3.1), we set $\lambda$ to 5, and $\epsilon$ is annealed from 0.1 to 0 over $w$=10M samples; for regularization

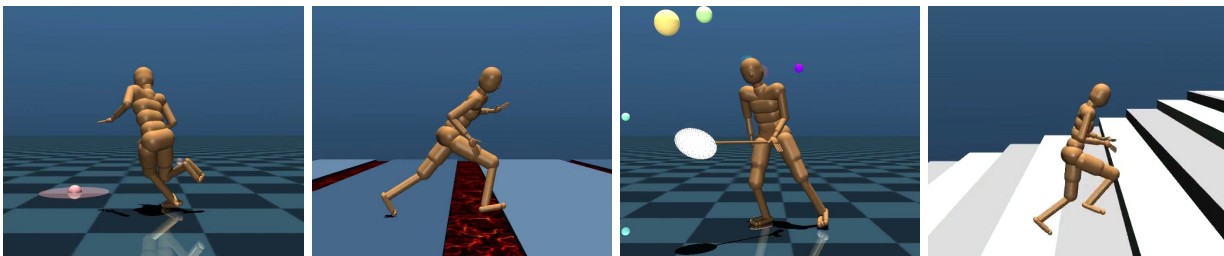

Figure 3: Trained agents acting in transfer tasks. From left to right: GoToTargets concerns locomotion along randomly sampled waypoints; in Gaps, the character is not allowed to touch the area between platforms; in Butterflies, floating spheres have to be caught with a light net; Stairs provides a series of ascending and descending stair-cases.

(3.2), we initialize $\delta$ with 0.01 and annealing to zero is performed over $w$=1M samples. The context ($c$) of previous high-level actions $z_{i<t}$ provided to $\pi_0(z_t|z_{i<t})$ is selected from $\{1, 4, 8\}$ for both exploration and regularization. When integrating priors as options (3.3), we evaluate option lengths of 2, 4 and 6 steps. Full training details are provided in Appendix C.

## 4.1 Pre-Training

Our pre-training stage yields several components: a latent space encoder and decoder, the latent space prior as well as a low-level policy. We first evaluate our pre-trained skill policy against the low-level policy described in CoMic (Hasenclever et al., 2020). Figure 4 compares the training reward achieved by both CoMic and our policy (z-prior). Learning proceeds slower in our setup, although both policies reach comparable performance at the end of training. We note that while we keep V-MPO hyper-parameters fixed across the z-prior and CoMic pre-training runs, the respective policy designs differ significantly. In CoMic, the high-level action space encodes both proprioceptive state and reference poses relative to the agent's state, while our policy is provided with VAE latent states and has to learn any relations to simulation states and rewards from experience.

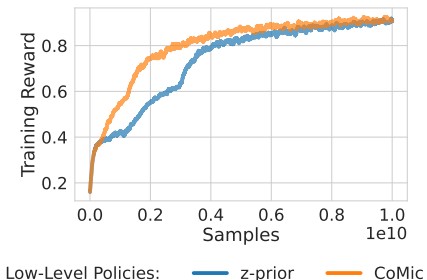

Figure 4: Training Reward during example-guided pre-training for our approach (z-prior) and CoMic. Both policies reach comparable final rewards.

Next, we investigate the interplay between the latent space prior $\pi_0$ and the low-level policy, which were trained independently but are grounded in a joint latent space. In Figure 6, we visualize different continuations sampled from the prior (colored) starting from a reference demonstration trajectory (silver). Qualitatively, we observe varied trajectories that frequently but not always resemble biologically plausible behavior; however, our VAE is not explicitly tuned towards realistic kinematic control. In many but not all cases, reenactment by the policy succeeds. For a quantitative analysis, we sample 100 latent state sequences of 10 seconds each from the prior. We then measure the average per-frame reward between reenacted and decoded states. The average per-frame reward on the original demonstration dataset is 0.89; on sampled trajectories, we observe a drop to 0.78.

## 4.2 Transfer Tasks

We plot learning curves of training runs on transfer tasks in Figure 5. Our results demonstrate faster learning and increased final performance when integrating latent space priors. In terms of general performance, exploration with the prior (z-prior Explore) provides consistent benefits; however, the nature of the task dictates the most useful integration method. For regularization, we find accelerated learning on Gaps and Butterflies, On GoToTargets, where observations allow the agent to plan ahead, utilizing the prior as an option policy results in fast learning. On Stairs, the prior helps a larger fraction of agents discover the second

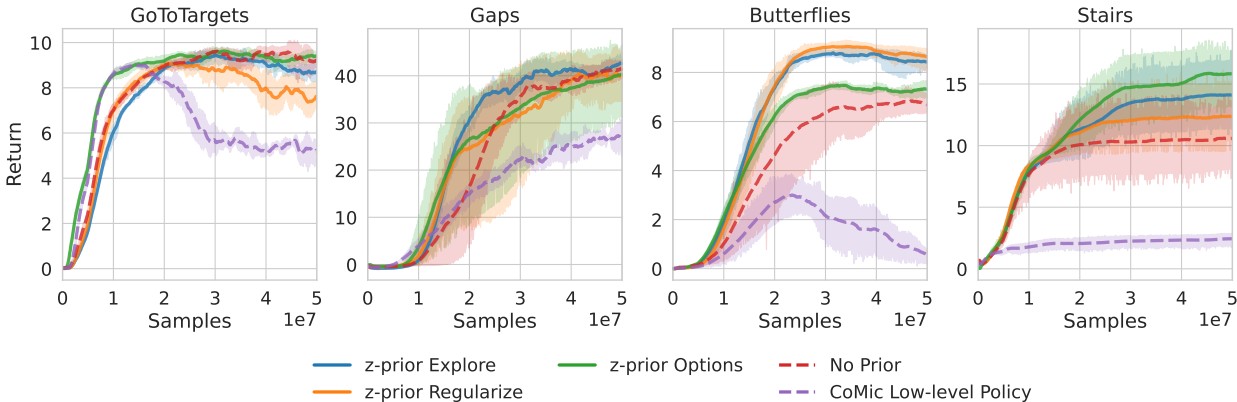

Figure 5: Learning curves on transfer tasks. Latent space priors (z-prior) accelerate learning and increase final performance, with the best method of integration depending on task characteristics. On GoToTargets, temporally extended options provide quick exploration; on Gaps and Butterflies, all variants with priors lead to faster exploration, with z-prior Explore and Regularize achieving the highest returns. On Stairs, utilizing the prior as an option policy leads to best mean performance. The comparison to CoMic validates our low-level policy (red). Results averaged over 4 runs, except for Stairs (10 runs).

staircase leading down from the platform, which results in a return above 10. Generally, forming options with $\pi_0$ restricts the actions that the high-level policy can perform, and, while useful for exploration, stifles final performance in the Gaps and Butterflies tasks.

Compared to CoMic, our low-level policy achieves superior performance on all tasks, even though CoMic often allows for faster learning early in training, e.g., on GoToTargets. We perform further comparisons of both low-level policies in Appendix D. To highlight the difficulty of our tasks, we also trained non-hierarchical policies from scratch with SAC and found that they did not make meaningful learning progress (Appendix H.1). Recently, Wagener et al. (2022) released an offline dataset for the CMU motion capture database. In experiments detailed in Appendix H.3, we observe that, with a comparable offline-learned policy, their high-level PPO agent fails on our benchmark tasks.

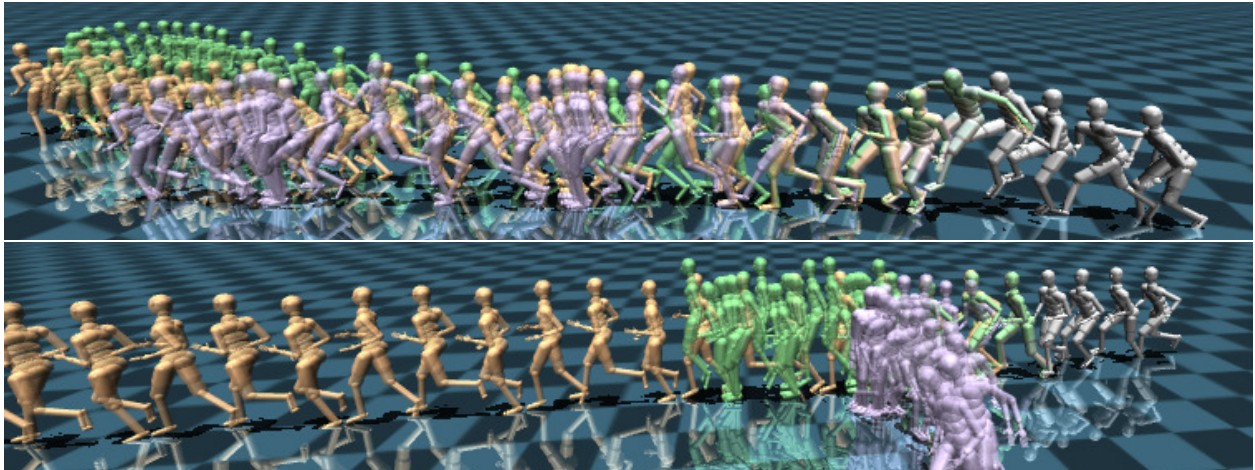

Figure 6: Sequences sampled from the latent space prior (colored), starting from a demonstration trajectory (silver) and decoded with the VAE. Top: the short jumping motion is completed, and sampled trajectories continue walking and running with changes in direction (orange and green) or include jumps from the training data (mauve). Bottom: the sampled trajectories include continuing the running motion (yellow) or changing walking speed after a turn (mauve).

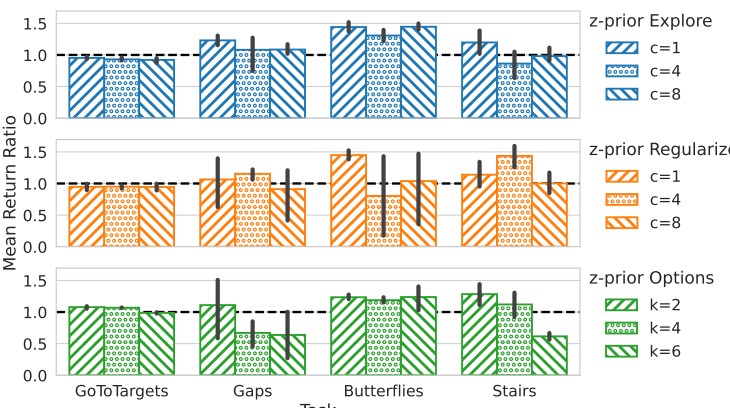

Figure 7: Variations of hyper-parameters when integrating priors. We plot the mean return achieved over the full course of training relative to learning without a prior, i.e., values larger above 1 signify improvements from latent space priors. Best configurations depend on the nature of the task at hand, with $c=1$ and $k=2$ providing good overall performance.

Overall, we find that optimal hyper-parameters such as option length or the number of past actions provided to the prior also depend on the concrete task at hand. For Figure 5, we select fixed hyper-parameters for all methods (explore $c$ (context) $=1$; regularize $c=1$; options $k=2$) and report performance for other values in Figure 7. The results indicate that providing a larger context of actions for exploration or regularization is helpful in specific scenarios only (e.g., z-prior Regularize with $c=4$ on Stairs). Increasing option lengths (z-prior Options) is also not effective on our downstream tasks when focusing on achieving high eventual performance, whereas initial learning can be accelerated on GoToTargets and Butterflies. We refer to Appendix E for full learning curves of these ablations as well as further discussion. In Appendix G, we estimate that good policies on our tasks act markedly different compared to the demonstration data, which supports the finding that providing larger context to our prior is generally not helpful.

### 4.3 Offline Datasets

We evaluate latent space priors on two offline RL datasets: maze navigation with a quadruped "Ant" robot and object manipulation with a kitchen robot (Fu et al., 2021). We train low-level policies and latent space priors on offline data, and then learn a high-level policy online in the downstream task. We obtained best results by training a state encoder jointly with the low-level policy, rather than learning a VAE in isolation (C.6). For AntMaze, we follow the experimental setup from OPAL (Ajay et al., 2021) and use their online RL variant as a baseline; for the kitchen robot, we compare to SPiRL (Pertsch et al., 2020). Both OPAL and SPiRL acquire latent space conditioned, temporally extended low-level primitives on a given offline dataset, along with a state-conditioned prior $\pi_0(z|s)$ to regularize high-level policy learning.

Results in Figure 8 show that latent space priors are effective in both domains. In the maze environment (Fig. 8, left), all forms of prior integration solve the task quickly, with options (we use k=10) yielding best results. Learning a high-level policy without a prior (no prior) requires about 3M samples to start reaching

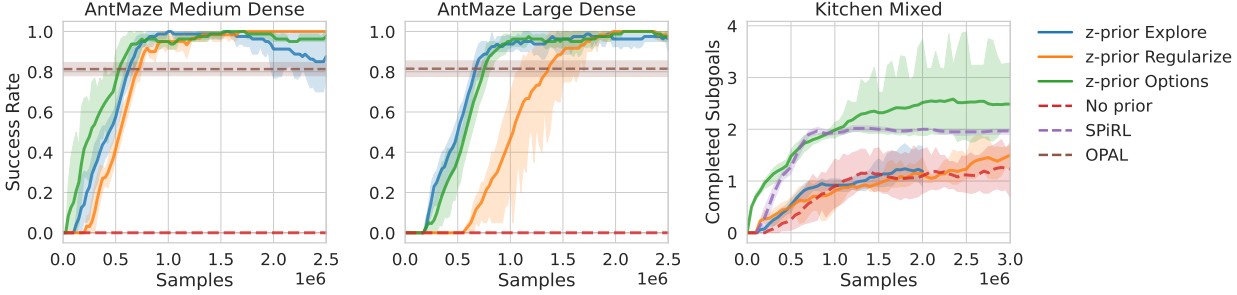

Figure 8: Results for pre-training low-level policies and latent space priors on offline datasets. OPAL results are provided by Ajay et al. (2021) at 2.5M samples for the medium and 17.5 samples for the large maze, respectively.

the goal position (not shown). We were not able to tackle the sparse-reward variant of this task successfully (OPAL achieves a success rate of 81.6% for the medium maze but fails on the large maze). We hypothesize that this is due to our priors not being conditioned on the current environment state; on the other hand, this allows us to easily transfer the pre-trained components to the large maze (Fig. 8, center). In the kitchen task, we obtain a higher average return than SPiRL with the option integration (k=10). Both exploration and regularization with latent space priors fail to significantly accelerate learning in this environment.

## 5 Related Work

Our work resides in a long line of research concerned with utilizing demonstration data to assist reinforcement learning agents, which, in their basic setting, attempt to solve each new task by training a policy from scratch via trial and error (an overview is provided by Ravichandar et al. (2020), among others). While demonstration data can be used to acquire policies, models, value or reward functions directly (Pomerleau, 1988; Atkeson and Schaal, 1997; Ng and Russell, 2000; Hester et al., 2018), we seek to acquire structures that accelerate learning in unseen transfer tasks. Further, an important distinction is the type of demonstrations that informed the design of a given method. In the batch or offline RL setting in particular, the dataset contains state-action trajectories and a corresponding reward function is assumed to be available (Lange et al., 2012; Ajay et al., 2021; Shiarlis et al., 2018; Levine et al., 2020). As demonstrated in Section 4.3, our approach is compatible with this setting, but our main focus is on demonstrations consisting of observations only, with the assumption that simulated environments are available for low-level policy learning.

**Skill Learning from Demonstrations** A paradigm that lends itself particularly well to transfer settings is the discovery of low-level behavior, or skills, to provide reusable abstractions over state or action spaces sequences (Dayan and Hinton, 1992; Thrun and Schwartz, 1995; Sutton et al., 1999; Eysenbach et al., 2019; Hasenclever et al., 2020; Gehring et al., 2021). Discovering skills from demonstrations has been the focus of a large body of work (Konidaris et al., 2012; Niekum et al., 2012; Paul et al., 2019). Among recent works utilizing deep neural networks as policies, Pertsch et al. (2020) and Ajay et al. (2021) propose to segment offline data into fixed-length sub-trajectories and obtain skills by auto-encoding action and state-action sequences, respectively, to latent states and back to actions. Singh et al. (2021) propose to learn an invertible action space transformation from action sequences which then forms a low-level policy. We refrain from introducing further inductive biases regarding the segmentation of demonstrations; instead, we model whole trajectories with a generative sequence model and train a single-step skill policy. Outside of the context of demonstrations, our method bears similarity with Co-Reyes et al. (2018), who also consider latent sequence generation models and subsequent execution via low-level policies. In their work, learning is performed directly on downstream tasks, and the high-level policy is implemented as a model-based planner. Evaluation is however limited to environments with comparably simple dynamics, and learning policies from scratch poses severe challenges in our tasks. Recently, Tirumala et al. (2020) formalized the notion of behavior priors, i.e., policies that capture behavior by regularization with a divergence term, and Siegel et al. (2020) addresses learning them from demonstrations presented as offline data.

**Generative Sequence Models** Our latent space priors are inspired by recent successes of generative sequence models, in particular transformers, in domains such as natural language, music or speech (Brown et al., 2020; Dhariwal et al., 2020; Tan et al., 2022). Investigations into utilizing transformers for decision-making have largely focused on the offline setting, where actions are available as ground-truth targets for generation (Chen et al., 2021; Janner et al., 2021; Reed et al., 2022). Recent studies also investigate online fine-tuning (Zheng et al., 2022) and model-based RL (Micheli et al., 2022). With respect to improving exploration, Zhang et al. (2022) employ recurrent networks as generative planners integrated into model-free policy learning. Their planning models take states and task rewards into account and are hence tailored to specific downstream tasks, whereas we develop general-purpose priors for a specific agent modality.

**Humanoid Control from Motion Capture Data** We evaluate our method on a simulated humanoid and motion capture data demonstrations, a setting that is also subject of a separate stream of research concerned with synthesizing natural motions of avatars for kinematic and physics-based control (Peng et al., 2018; Ling et al., 2020). While our latent space prior provides opportunities for kinematic control, we focus on

transfer performance in physics-based, sparse-reward tasks. Deep RL techniques have been shown to perform successful imitation learning of motion capture demonstrations via example-guided learning (Peng et al., 2018; Won et al., 2020; Wang et al., 2020), and several works investigate low-level skill learning and transfer in this context (Hasenclever et al., 2020; Won et al., 2020; Liu et al., 2021). Our pre-training method closely resembles CoMic (Hasenclever et al., 2020), which learns a latent high-level action space and a low-level policy in an end-to-end fashion; crucially, we separate these steps and further leverage the temporal structure in demonstrations with latent space priors. Won et al. (2021) target multi-agent downstream tasks, as do Liu et al. (2021), where skills learned from motion capture data serve as priors for new policies trained from scratch in the environment's native action spaces. We regard these works as complementary to our approach and focus on transfer settings to single-agent downstream tasks by training individual high-level policies. With regards to priors for high-level learning in Humanoid control, Peng et al. (2021) employ a learned discriminator to achieve behavior of a specific style. Bohez et al. (2022) consider transfer of policies to real robots and extend CoMic with an auto-regressive prior on downstream tasks. They show that correctly scaled auto-regressive regularization improves energy use and smoothness during locomotion.

## 6 Conclusion

We introduced a new framework for leveraging demonstration data in reinforcement learning that separates low-level skill acquisition and trajectory modeling. Our proposed latent space priors provide various means to accelerate learning in downstream tasks, and we verified their efficacy in challenging sparse-reward environments. Our experimental results also indicate that, among the methods for utilizing latent space priors that we investigated, the nature of downstream tasks determines their ultimate utility, such as quick exploration or increased final performance. Such trade-offs appear naturally in RL applications, and we regard the interplay of prior utilization and high-level policy learning as a promising avenue for future work.

Our approach is aimed at settings that benefit from the availability of proprioceptive motor skills, which is a common case in robotics applications. The flexibility of latent space priors makes extensions towards classes of problems in which broader repertoires of skills are required, e.g., object manipulation, attractive subjects of further study.

**Acknowledgements:** We thank Josh Merel and Alessandro Lazaric for insightful and helpful discussions.

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

## A    Demonstration Data

Our demonstration dataset originates from the CMU Graphics Lab Motion Capture Database [3], which we access through the AMASS corpus (Mahmood et al., 2019). We select a subset of the "locomotion" corpus from Hasenclever et al. (2020), containing about 20 minutes of various walking, running, jumping and turning motions. Each AMASS motion consists of 3D joint rotations and a global position for each frame, as well as parameters that define the body shape (Loper et al., 2015). We pre-process all motions with fairmotion (Gopinath and Won, 2020) as follows. First, we resample each motion so that its framerate matches the control frequency of the imitation and downstream task environments, i.e., 33.3Hz. Second, we determine custom SMPL shape parameters that approximate our simulated humanoid. The resulting skeleton is employed to compute forward kinematics for the representation learning loss, and to adjust the vertical position of the motion.

For representation learning (2.2), we auto-encode local joint angles as well as body position and orientation. The body's X and Y positions are featurized as velocities in X and Y direction to maintain location invariance. Body orientation and local joint rotations are featurized as forward and upward vectors (Zhang et al., 2018). We do not model joint velocities.

For low-level policy training via example-guided learning, we use the MuJoCo physics simulator (Todorov et al., 2012) and the "V2020" humanoid robot from dm_control (Tassa et al., 2020). We modify the robot definition slightly to ease conversion from the SMPL skeleton by adjusting the default femur positions and left shoulder orientation. SMPL poses are mapped to V2020 joint rotations with a manually defined mapping; notably, the MuJoCo model uses one-dimensional hinge joints only so that, e.g., for the elbow joints, two degrees of freedom are lost. We exclude head and finger joints from the conversion. Global and local joint velocities are computed with finite differences following Merel et al. (2019a).

## B    Transfer Tasks

The **GoToTargets** task is similar to the "go-to-target" task in Hasenclever et al. (2020); however, the version we consider requires a larger variety of locomotion skills (fast turns and movements). Instead of selecting the first goal randomly and the following goals in the vicinity of the first, we sample all goals randomly within an area of 8x8 meters. For the minimum distance to obtain a reward of 1, we use 0.5m instead of 1.0m. A new goal is provided as soon as the robot achieves the minimum distance, rather then switching to a new goal after 10 steps as in Hasenclever et al. (2020). Hence, agents naturally achieve lower returns in our task. The agent observes the relative positions of the current and next goal.

In the **Butterflies** task, a light "net" is fixed to the left hand of the robot. Spheres are randomly placed within a 4x4m area, at heights ranging from 1.43m to 2.73m. The agent receives a reward of 1 if the net touches a sphere. We limit the number of spheres to 10, which denotes the maximum return. Untouched spheres are encoded as observations by projecting their location onto a unit sphere placed at the robot's head location. The unit sphere is rasterized with longitudes and latitudes spanning 10 degrees each, and each entry is set to the exponential of the negative distance to the respective closest sphere, or to zero if no sphere is projected onto that location.

The **Gaps** and **Stairs** tasks have been introduced by Gehring et al. (2021); we scale stairs, gap and platform lengths by 130% to better fit the humanoid robot we use in this study, and do not end the episode or provide negative rewards if the robot falls over. For **Gaps**, the agent receives a reward of 1 when the robot reaches a platform; if the robot touches the lower area between two platforms, the episode ends immediately with a reward of -1. In the **Stairs** task, a reward of 1 is obtained for each new stair that the humanoid's body reaches.

---

[3]http://mocap.cs.cmu.edu/

## C    Training Details

### C.1    Auto-Encoding Demonstration States

Our convolutional VAE consists of encoder and decoder residual networks (Dhariwal et al., 2020). All convolutional layers are one-dimensional and use a kernel size of 3 with 64 output channels. For the encoder, a single convolution layer is followed by 4 residual blocks and another final convolution with 32 output channels. From the 32-dimensional encoder output, we predict the means and standard variation of an isotropic normal distribution which is KL-regularized towards $\mathcal{N}(0, 1)$. The KL term is weighted with a factor of 0.2. The decoder mirrors the encoder's network structure, with the last layer consisting of a transposed convolution. A final output layer performs a convolution and restores the input dimensionality.

VAE inputs and outputs are the body's X and Y velocities, Z position, global orientation and all local joint rotations. X and Y velocities are integrated to positions relative to the start of the mini-batch, and global joint locations and rotations are computed by propagating local joint rotations through the skeleton (App. A, Villegas et al. (2018)). The reconstruction loss resembles the reward function proposed by Hasenclever et al. (2020), albeit without scoring velocities. In the following, P denotes joint locations, R denotes 6D rotations (Zhang et al., 2018) and Q denotes quaternions computed from R excluding the root joint:

$$\mathcal{L}_{\text{recons}} = -\frac{1}{2}\left(1 - \frac{1}{0.6}\left(d_{\text{jpos}} + d_{\text{jrot}}\right) + \left(0.1 r_{\text{root}} + 0.15 r_{\text{app}} + 0.65 r_{\text{bq}}\right)\right)$$
$$d_{\text{jpos}} = ||P^{\text{ref}} - P||_1$$
$$d_{\text{jrot}} = ||R^{\text{ref}} - R||_1$$
$$r_{\text{root}} = \exp\left(-10||P^{\text{ref}}_{\text{root}} - P_{\text{root}}||^2\right)$$
$$r_{\text{app}} = \exp\left(-40||P^{\text{ref}}_{\text{app}} - P_{\text{app}}||^2\right)$$
$$r_{\text{bq}} = \exp\left(-2||Q^{\text{ref}} \ominus Q||^2\right)$$

$Q^{\text{ref}} \ominus Q$ refers to the quaternion difference. Appendages (app) consist of the head, left and right wrist and left and right toe joints of the SMPL skeleton.

We train the VAE on mini-batches of 128 trajectories with a window length of 256. Parameters are optimized with Adam (Kingma and Ba, 2015) using a learning rate of 0.0003 with a linear warmup over 100 updates. We perform 100000 gradient steps, shuffling the training data between epochs.

### C.2    Modeling Trajectories

We use the Transformer implementation from Dhariwal et al. (2020) to train an auto-regressive generative model on encoded demonstration trajectories. Inputs are mapped to 256-dimensional embeddings, and the Transformer consists of 4 blocks, each computing single-head, dense, causal attention followed by layer normalization (Ba et al., 2016), a two-layer MLP with GeLU activation and another layer normalization. From the Transformer output, we predict mean and standard deviation for each of the 8 mixture components as well as logits for mixture weights. Entropy regularization is performed by sampling from the output distribution and subtracting the resulting log-probabilities to the loss term with a factor of 0.2.

Training is performed over 1M gradient steps, with mini-batches of 32 windowed trajectories of length 64. Optimization settings match VAE training, i.e., Adam with a learning rate 0.0003 and warmup over 100 updates.

We apply tanh activations to encoded states and sequence model outputs to improve compatibility with the Soft Actor-Critic algorithm used on downstream tasks. Preliminary experiments regarding the application of SAC to unsquashed action spaces demanded additional regularization of policy output layers, which inhibited learning progress.

### C.3 Low-Level Policy

We perform example-guided training of low-level policies as described in Hasenclever et al. (2020). We implement V-MPO in a distributed, synchronous setting with 320 actors, and perform 100 gradient steps, each on 160 trajectories of length 32, after collecting the required number of samples. For parameter optimization, we use Adam with a learning rate of 0.0001. Policy and value networks use 4 hidden layers with skip connections to input features (Sinha et al., 2020), each containing 1024 units, and the encoder architecture of our CoMic baseline matches the policy network. The value network predicts each reward term with a separate output unit. All networks process their inputs with layer normalization and We train each low-level policy with 10B samples (about 2M gradient steps).

We found that when training our low-level policy on static references, i.e., future input frames that are not featurized relative to the robot's current pose, it failed to correctly imitate longer clips. This was due to missing guidance regarding the X and Y positions, as our latent states encode X and Y velocities only without taking the current simulation state into account. We counteract this in two ways: first, we supply policy and value networks with two additional inputs specifying the X and Y offset to the current reference pose (while latent states encode immediate future reference poses). These two inputs are set to zero when employing the policy on downstream tasks. Second, we provide the value function with relative root body positions of future poses.

For CoMic low-level policy training, we follow Hasenclever et al. (2020) in providing 5 future reference poses to the encoder, and regularizing its output distribution towards $\mathcal{N}(0, 1)$ with a factor of 0.0005. In preliminary experiments, we were not able to obtain well-performing CoMic policies on static reference features.

### C.4 High-level Policies

On transfer tasks, we freeze the low-level policy and train high-level policies operating in the learned latent space. High-level policy networks consist of 4 hidden layers of 256 units each, with skip connections to input features. We employ SAC with the following hyper-parameters:

| Parameter | Value |
| --- | --- |
| Learning rate | 0.0001 |
| Batch size | 256 |
| Replay buffer size | 5M |
| Samples between gradient steps | 300 |
| Gradient steps | 50 |
| Warmup samples | 10000 |
| Initial temperature coefficient | 1 |
| Number of actors | 5 |

During warmup, we perform exploration by randomly sampling actions within the high-level action space and do not perform parameter updates. Other hyper-parameters, with the exception of network architectures, are chosen to match Haarnoja et al. (2018).

We select above hyper-parameters by training high-level policies for our low-level policy as well as CoMic for 20M frames on the GoToTargets and a separate hurdles jumping task; sweeps were performed over batch sizes in $\{256, 512\}$, samples between gradient steps in $\{50, 100, 200, 300, 500\}$, replay buffer size in $\{2M, 5M\}$, learning rates in $\{0.0001, 0.0003\}$ and initial temperature coefficients in $\{0.1, 1\}$.

Evaluations are performed every 100k samples on 50 task instances using fixed, per-instance seeds.

### C.5 Compute Infrastructure

We implement models and training algorithms in PyTorch (Paszke et al., 2019). For pre-training, parallel environment execution and feature computation are implemented in C++. We train demonstration auto-encoders on 2 GPUs and sequence models on a single GPU. For low-level policy training, we use 16 NVidia

V100 GPUs and 160 CPUs; training on 10B samples takes approximately 5 days. High-level policies are trained on a single GPU and require 2-3 days to reach 40M samples.

### C.6 Offline Datasets

Experiments in Section 4.3 were performed on offline datasets from D4RL[4]. For AntMaze experiments, we pretrain low-level policies and latent space priors on the dataset from "antmaze-medium-diverse-v1", and learn high-level policies in the "antmaze-medium-diverse-v0" and "antmaze-large-diverse-v0" environments, providing a dense reward of $-||g - ant_{xy}||$. For pre-training, we exclude the agent's X and Y coordinates in accordance with (Ajay et al., 2021) (personal communication). The kitchen experiments are performed on "kitchen-mixed-v0", which is only partially observable (joint velocities are not included in the observation). We combat this by presenting the current as well as the last three observations to all policies.

We found that best results are obtained by jointly training a latent space encoder and low-level policy, i.e., we do not train a full VAE to perform state reconstruction. Instead, an encoder $q$ (parameterized as the low-level policy) computes the mean and standard deviation of an isotropic normal distribution, subject to KL regularization, from a concatenation of the next five observations. Latent states are sampled from the resulting distribution and provided to the low-level policy, which is trained with behavior cloning with the loss

$$\mathcal{L}(\phi, \psi) = - \mathop{\mathbb{E}}_{\substack{(s_t, a_t) \sim D \\ z_t \sim q_\phi(s_{t+1}, ..., s_{t+5})}} ||\pi_{\text{lo}, \psi}(s_t, z_t) - a_t||$$

For both domains, policies consist of 4 hidden layers of 256 units each, with skip connections to network inputs. Latent space dimensions are adopted from Ajay et al. (2021) and Pertsch et al. (2020) with 8 and 10 for AntMaze and Kitchen environments, respectively. We train low-level polices with 0.2M update steps, using a batch size of 256, Adam with learning rate $3e^{-4}$ and a KL regularization coefficient of $1e^{-3}$. Latent space priors consist are implemented as transformers with 2 layers of 256 units each, and output a Gaussian mixture distribution with 4 components. Training data is presented as sequences of length 64, and we perform 1M gradient steps with Adam and a learning rate of $3e^{-4}$ as well as entropy regularization with a factor of 0.01.

For high-level policy learning, we use SAC as in Appendix C.4 but with a learning rate of $3e^{-4}$, a replay buffer size of 1M, 50 samples between gradient steps, 5000 warmup samples and a single actor only. Latent space prior integration is done with the following hyper-parameters:

| Method | Parameter | AntMaze | Kitchen |
|---|---|---|---|
| z-prior Explore | Prior steps $\lambda$ | 16 | 5 |
| | Prior context $c$ | 4 | 4 |
| | Prior temperature $\epsilon$ | 1.0 | 0.0 |
| | Prior rate $\epsilon$ | 0.01 | 0.01 |
| | Annealing steps | $\infty$ | 1M |
| z-prior Regularize | Regularization factor $\delta$ | 0.5 | 0.01 |
| | Prior context $c$ | 1 | 4 |
| | Annealing steps | 2M | 0.5M |
| z-prior Options | Option length $k$ | 10 | 10 |

## D  Low-Level Policy Comparison

In Section 4.2, high-level policies utilizing a CoMic low-level policy (Hasenclever et al., 2020) exhibit faster initial learning but achieve lower final performance compared to our low-level policy. We investigate the behavior of both low-level policies in additional experiments.

First, we plot random walks with both low-level policies in the GoToTargets task in Figure 9. For 10 episodes, we uniformly sample random high-level actions. With the CoMic low-level policy, this results in

---

[4]We use the version from SPiRL at `https://github.com/kpertsch/d4rl/tree/a4e37c6`

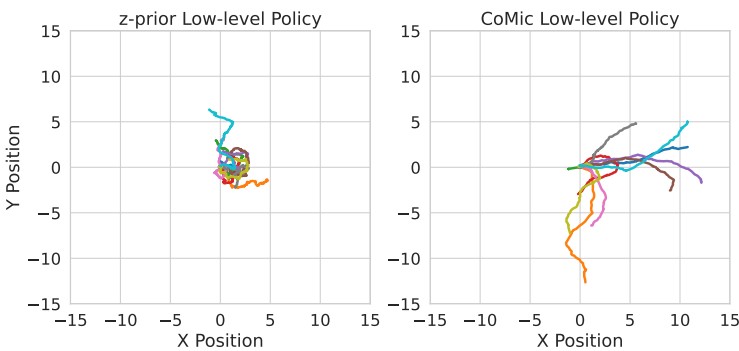

Figure 9: Random walks with our low-level policy and CoMic in the GoToTargets task.

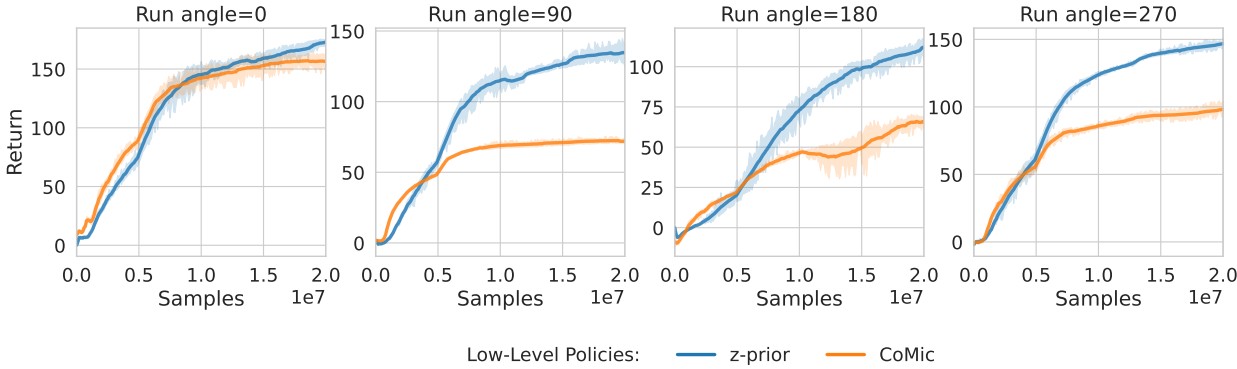

Figure 10: Low-level policy evaluation on a simple locomotion task, without using latent space priors. A dense reward is provided for moving the robot's body at a specific angle. The demonstration data contains clips running forward (angle=0) only; for other directions, our low-level policy (z-prior) achieves substantially higher rewards.

basic locomotion behavior, i.e., taking single steps in different directions, while our low-level restricts the random walk to a comparably small region. This matches the observation by Hasenclever et al. (2020), which they suggest leads to faster learning.

To better understand the gap in final performance between the two policies, we use them in a directed locomotion task. Here, the reward is provided by the length of the current hip translation projected onto a unit vector at a specified angle. For example, an angle of 180 degrees requires the humanoid to walk backwards, and an angle of 90 degrees requires a sideways movement. The results in Figure 10 indicate that our low-level policy (z-prior) outperforms CoMic in all variants but forward running (angle=0). In the demonstration data, running motions are generally forward-directed, and only a single clip contains a backwards walking motion. The findings suggest that the CoMic low-level policy provides behavior close to the demonstration data at the expense of restricting high-level policy control.

To further support the hypothesis on restricted high-level controllability with CoMic, we select two states encountered during downstream task training and compare how low-level actions are influenced by high-level actions. In Figure 11, we plot the cosine similarity for pairs of 256 random high-level actions (X-axis) and their resulting native actions (Y axis). Low-level actions produced with CoMic have a higher overall similarity, in particular in the stranding pose (left), which implies reduced controllability compared to the z-prior low-level policy.

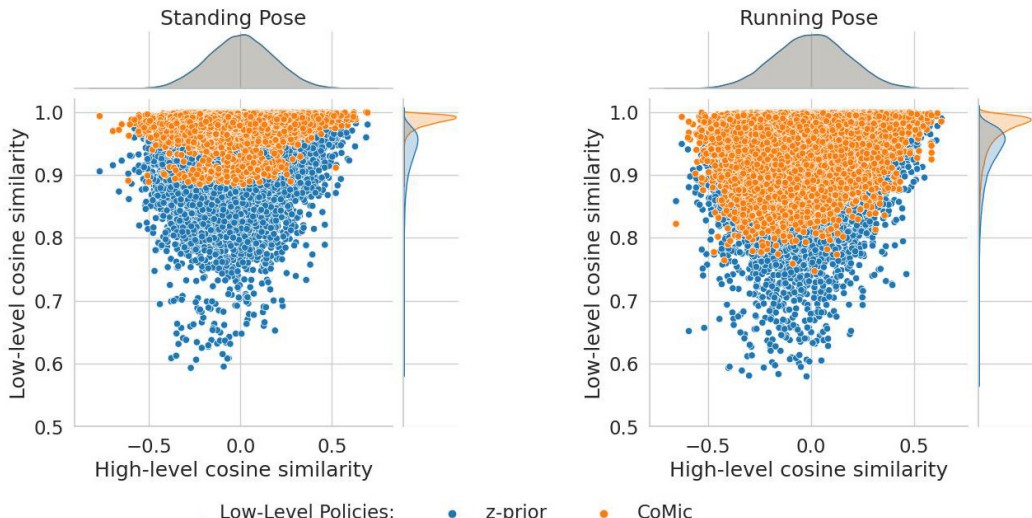

Figure 11: Comparison of cosine similarity between random high-level actions and the similarity of the resulting low-level actions in two different states. Our low-level policy (blue) provides a larger range of low-level actions, while CoMic (orange) produce more similar low-level actions, indicating reduced controllability.

## E  Ablation Studies

In Figure 12, we plot learning curves for several ablations regarding hyper-parameters for integrating latent space priors in high-level policy learning.

When using the prior for exploration (top row), providing a longer context to the prior does not have a significant effect in GoToTargets, which matches the demonstration data well, but hurts learning speed (Gaps, Butterflies) and final performance (Stairs) in other tasks. We hypothesize that a longer context of high-level policy actions which are out-of-distribution for the prior renders its generated sequences less helpful for exploration.

If the prior is used as a regularization target (middle rows), annealing is performed over a relatively short window at the start of training. The ablation in the third row shows that regularization conflicts with learning the task at hand: in GoToTargets and Stairs, where the learning starts early, annealing over 5M frames clearly delays learning progress even though later performance is slightly increased. Thus, while regularizing towards the latent space prior yields good initializations, the prior's ignorance of ground-truth simulation states and rewards makes it less useful for long-term guidance.

In the bottom row of Figure 12, we vary the option length when generating temporally extended actions with the prior. Longer options result in a loss of performance for Gaps and Stairs; in GoToTargets, for which demonstration clips provide the necessary basic skills of accelerating and turning, we see faster initial learning with an option length of 4.

## F  Planning with Latent Space Priors

We perform a brief investigation into planning with latent space priors on a simple navigation task with a dense, distance-based reward function. A goal is randomly placed on circle with a radius of 4m, and the a trial is considered successful if the robot was moved within a distance of 0.5m to the goal position. Rather than learning a high-level policy, we sample candidate sequences from the learned prior in order to steer the low-level policy. Planning is performed in a model predictive control fashion: we first generate a number of latent space sequences, decode them into robot poses with the VAE (Section 2.2) and select the best-performing sequence to condition the low-level policy for the next steps (Algorithm 1). We score sequences by determining the maximum reward reached among any of decoded states.

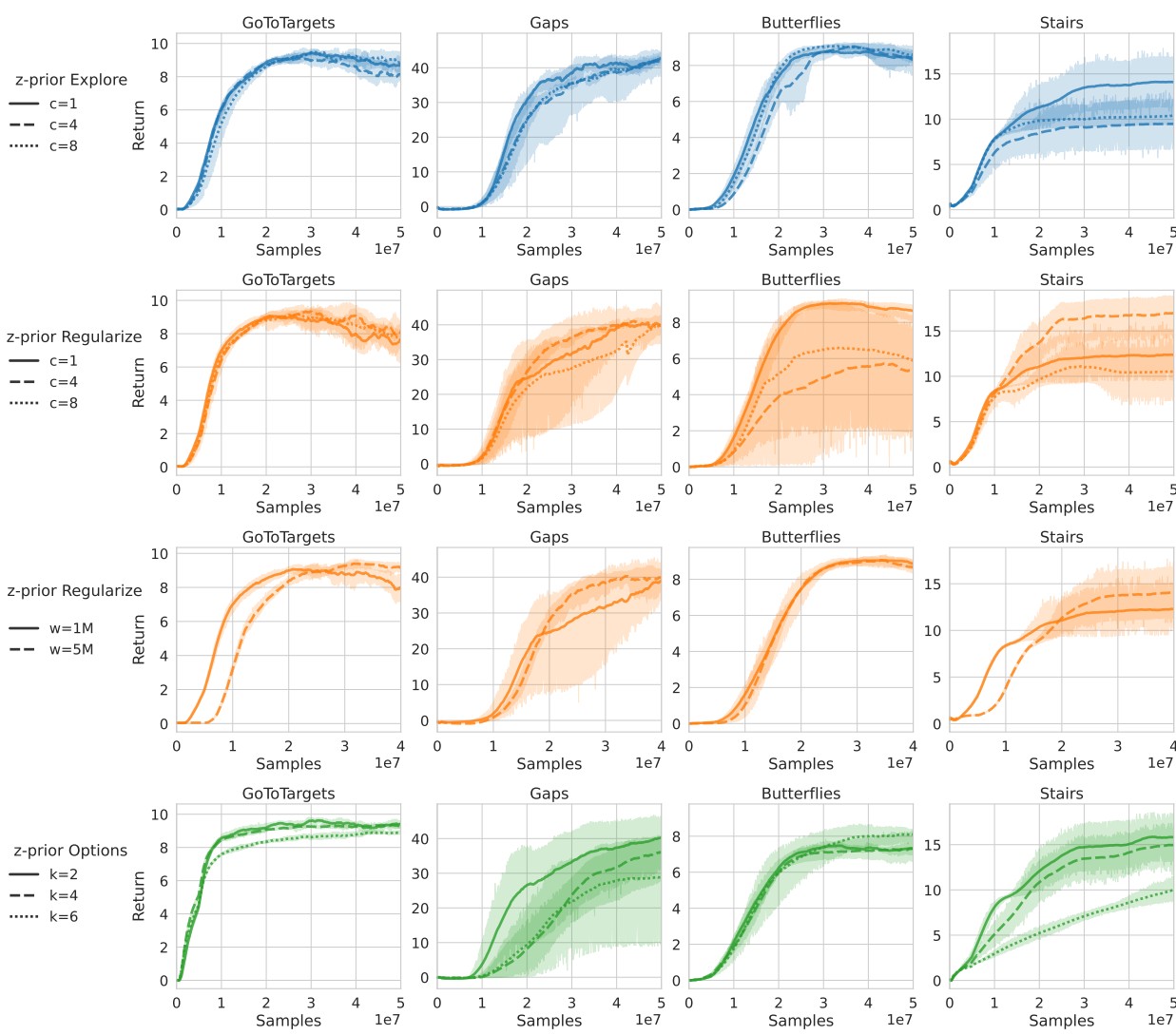

Figure 12: Learning curves for hyper-parameter ablations when integrating latent space priors in high-level policy learning.

---

**Algorithm 1** Model Predictive Control with Latent Space Priors

---

**Require:** Latent space prior $\pi_0$, low-level policy $\pi_{\mathrm{lo}}$, VAE encoder $q$ and decoder $p$
**Require:** Horizon $h$, Planning interval $i$
**Require:** Reward function $r : \mathcal{S} \to \mathbb{R}$
 1: states $\leftarrow$ [initial state $s$]
 2: **while** not done **do**
 3:     $\mathbf{z} \leftarrow q(\text{states})$
 4:     $\mathbf{z}^* \leftarrow \arg\max_{z_1',\dots,z_h' \sim \pi_0(z_1',\dots,z_h'|\mathbf{z})} \max_{s' \in p(z_1',\dots,z_h')} r(s')$
 5:     **for** $t = 1, \dots, i$ **do**
 6:         $a \leftarrow \mathbb{E}_a[\pi_{\mathrm{lo}}(a|s, z_t^*)]$
 7:         $s \leftarrow \mathrm{STEP}(a)$
 8:         states $\leftarrow$ [states, $s$]
 9:     **end for**
10: **end while**

---

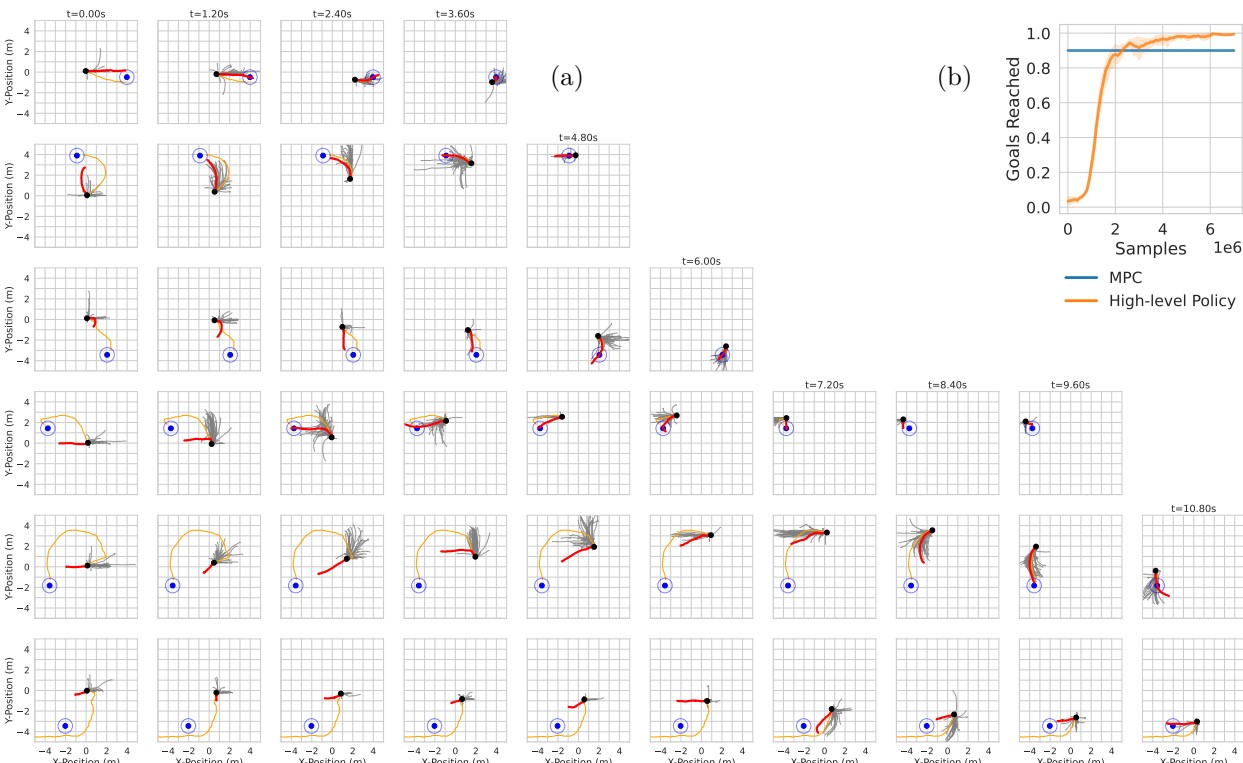

Figure 13: (a) Visualization of rollouts when planning with latent space priors, projected onto the X/Y plane. The robot (black dot) initially faces the positive X direction and is tasked to navigate to a goal marked in blue. A subset of candidate plans is plotted with grey lines; the selected plan is highlighted in red. The orange line depicts the actual future trajectory achieved. (b) Comparison to learning progress of a high-level policy on this task.

We re-plan every 4 environment steps by sampling 1024 candidate sequences of length 64. The variance of trajectories is increased with a high temperature for sampling (we scale the logits of the mixture distribution by a factor of 9 and the standard deviation of mixture components by 3). We evaluate on 50 randomly sampled goals, and compare the MPC implementation against the learning curve of a high-level policy that is trained on randomly sampled goals (Figure 13(b)). With MPC, we reach 90% of the sampled goals within the time limit of 1000 steps (30s), and 68% within 500 steps. A high-level policy with our default settings requires about 2.5 million samples to successfully reach the same fraction of goals. The learned policy reaches goals much faster, but often reaches them stumbling and falling (the episode ends when the goal is reached).

We visualize the planning process for several trials in Figure 13(a). While the sampled and decoded rollouts provide sufficient variety and quality to obtain useful plans (red lines), the low-level policy is not able to reenact them accurately in all cases. In particular, likely due to the demonstration data we considered, following turns from sampled trajectories is challenging[5]. This mismatch can be alleviated by frequent re-planning. The bottom row in Figure 13(a) depicts a failed trial (later steps not shown): the character misses the goal, as indicated by the orange line showing the future trajectory, and the low-level policy is not able to follow future plans to turn around.

## G Coverage of Demonstration Data

In a best-case scenario, the demonstration dataset would cover all behavior required on transfer tasks. In our setup, the final policies that were found to perform well on transfer tasks are still markedly different

---

[5]A learned high-level policy can however explore the full latent action space and discover suitable actions to perform fast turns in the GoToTargets task.

from the original demonstration data. We quantify this by evaluating the likelihood of evaluation rollouts under our latent space prior. The following data was obtained with 50 rollouts each from the best-performing z-prior Explore policy on each task and results in lower log-probabilities (averaged over sequence elements) under the prior. Note that rather than evaluating high-level action sequences, we encode the full motion produced during each episode with the VAE and score the resulting latent state sequence. We observe lower log-probabilities of final behaviors compared to the demonstration dataset.

| Sequences | Prior log-probability |
|---|---|
| Demonstrations | -134.8 ± 50.5 |
| GoToTargets | -195.8 ± 68.9 |
| Gaps | -375.8 ± 46.1 |
| Butterflies | -150.0 ± 32.7 |
| Stairs | -238.9 ± 102.8 |

## H  Additional Baselines

### H.1  SAC and Action Repeat

In Figure 14, we provide learning curves for further baselines: Soft Actor-Critic without a pre-trained low-level policy, and a high-level policy that repeats every action twice. SAC does not exhibit any learning progress, which highlights the utility of pre-trained skills for tackling our sparse-reward tasks. The action repeat baseline uses the same Q-function update as for integrating latent space priors via options (3.3). It achieves good results on GoToTargets and Butterflies which are only slightly lower compared to using the prior to generate options. In the Gaps task, however, no meaningful learning progress is made within the first 20M samples.

### H.2  LSTM and Non-parameteric Prior

In our main experiments, we utilized Transformers to implement latent space priors. In Figure 15, we show for the z-prior Explore setting that LSTMs (Hochreiter and Schmidhuber, 1997) (in a comparable configuration to our Transformer model) could also be employed, albeit with a reduction in performance on Gaps and Stairs. Selecting random sequences from the demonstration data rather than sampling from a sequence model can also work, matching the Transformer prior in 3 out of 4 tasks. This result matches well with our finding that providing a larger amount of context to the Transformer model is generally not helpful (Fig. 7). On the other hand, directly sampling from demonstrations requires the full (encoded) dataset to be available and cannot be utilized for "z-prior Options" and "z-prior Regularize", which achieve superior performance in GoToTargets and Stairs (Fig. 5).

### H.3  Low-level Policy from Offline Data

Recently, Wagener et al. (2022) released a large corpus (MoCapAct) for offline RL for the CMU motion capture database, with actions from expert policies trained to imitate individual clips. We select the same subset of clips that we use in our main experiments and train a multi-clip policy following the "locomotion-small" setup, i.e., with a reference encoder output dimensionality of 20. We then train a high-level policy as described in Wagener et al. (2022) on the transfer tasks we consider. Our robot model differs slightly from Wagener et al. (2022) and we hence adapt our environments for compatibility. In Figure 16, we plot learning curves for setups and observe that for MoCapAct, no learning progress is made. Doubling the sample budget for training for MoCapAct has no effect on our tasks (not shown). We note that different from Wagener et al. (2022), our transfer tasks are not initialized from an demonstration pose and do not employ early episode termination if the robot loses balance. To control for our custom environments, we re-implement the VelocityControl task with dense rewards and early termination and obtain learning progress with MoCapAct comparable to the reported results in Wagener et al. (2022) (Fig. 16, right).

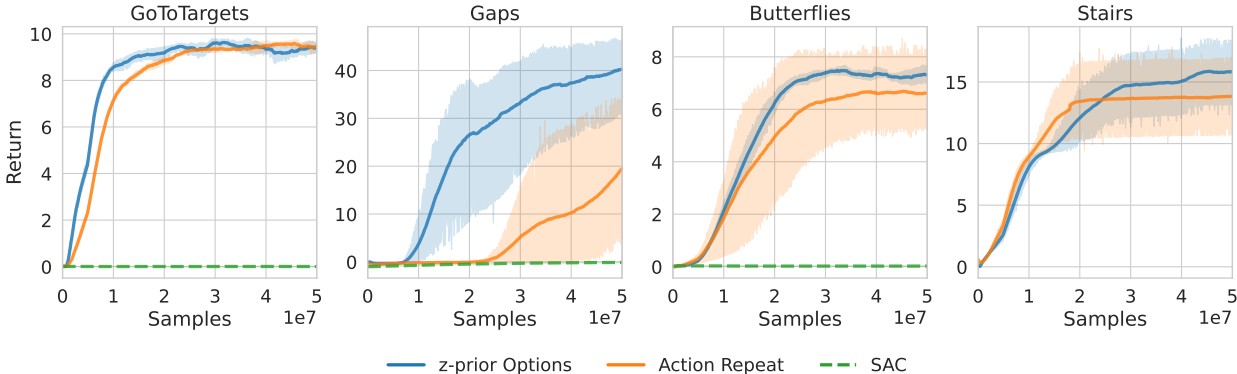

Figure 14: Further baselines: repeating high-level actions twice and plain Soft Actor-Critic.

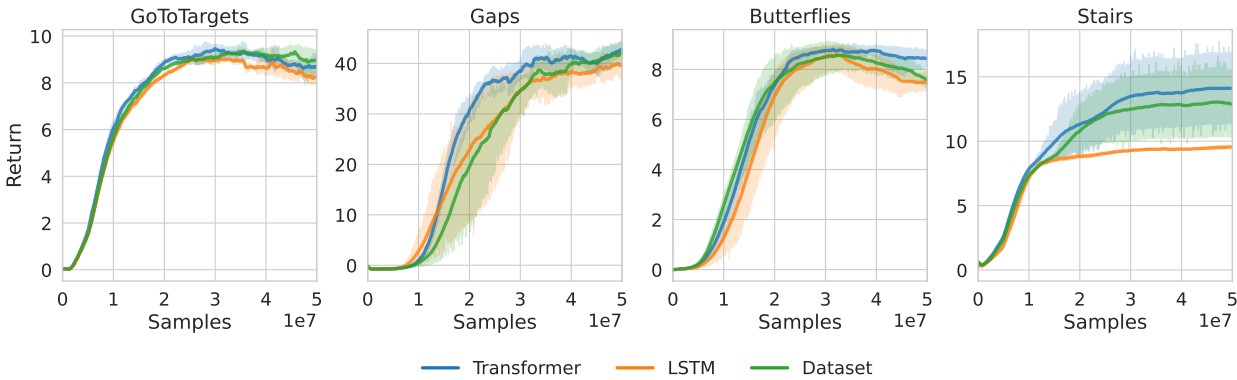

Figure 15: z-prior Explore with Transformer and a LSTM for the latent space prior. We also compare to sampling random latent state sequences from the demonstration dataset for exploration (Dataset).

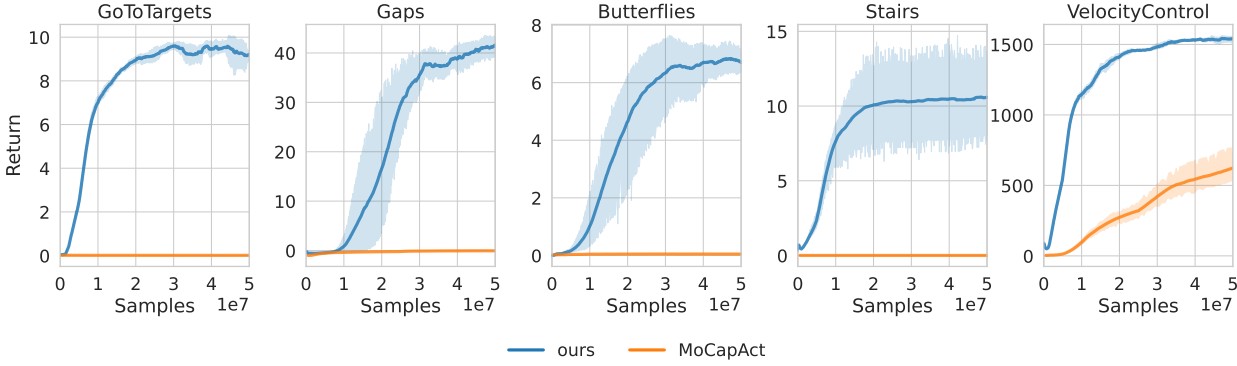

Figure 16: Comparing our training setup (without prior) to MoCapAct. The same demonstration dataset is used in both cases.

