# OpenReview forum: "Leveraging Demonstrations with Latent Space Priors"
_TMLR — Accepted by TMLR_

### Review · Reviewer_TeiQ · 2022-11-06

**Summary Of Contributions:**

The paper introduces a hierarchical decision making framework that helps make online reinforcement learning faster by using offline demonstration datasets. The high-level idea is to learn a probabilistic latent space of short-horizon skills from demonstrations and use the skill latent space in lieu of low-level action space to learn hard-exploration problems using online RL. Specifically, the paper proposes to train a latent-variable model (VAEs) of short-horizon behaviors from offline demonstration. The latent-space priors are then used to condition control policies to learn the desired behavior through example-guided RL without assuming access to ground-truth actions. The paper proposes to use the behavior priors in three different ways: (1) train an autoregressive model over the priors to generate meaningful exploration behaviors (2) regularize an RL learner to generate “prior-like” behaviors and (3) use the prior-conditional policies as options to learn a hierarchical RL policy.  The paper also shows that the latent-space priors can be used to perform planning. The method is evaluated on four simulated domains with simulated humanoid robot figures. The main result is that the prior-conditional low-level policies can indeed be used to aid task learning in the three different ways discussed above. And that all strategy discussed above can outperform a flat policy baseline (Hasenclever et al., 2020)


**Audience:**

Yes

**Broader Impact Concerns:**

The method presents no ethical implication as far as I can tell.

**Claims And Evidence:**

No

**Requested Changes:**

Overall I believe the authors need to spend more effort on showing why the presented method is important. There are many ways to show this, including but not limited to:

1. Present a clear, unified theoretical picture of the hierarchical decision making framework based on latent-space behavior prior. And show the empirical findings agrees with the theory.
2. Show that the proposed method outperforms a number of state-of-art methods on challenging simulated domain.
3. Show that the method works on an application of broad interest, e.g., significantly speed up the policy learning on a physical robot.

**Strengths And Weaknesses:**

Strengths and weaknesses:

Overall I enjoyed reading the paper — the motivation is stated clearly and the three strategies to make use of the learned latent space prior are easy to understand. I also like the spirit of presenting a variety of ways that one may approach hierarchical policy integration. Finally, the tasks, although all simulated, seem to be decently challenging.

My main criticisms of the paper focus on the following aspect (1) technical novelty and (2) insufficient baselines.

Like mentioned in the “strengths” part of the comment, I like the spirit of a unified view of hierarchical policy integration. However, I struggled to differentiate the proposed method from many other works published in the last few years. For example, as stated in the related works section, Co-Reyes et al. 2018, learns latent-space policy but uses only for planning. Pertsch et al. 2020, learns latent-space policy but assumes actions are available. Some other examples include Lynch et al. 2018 (Learning from Play Data), which learns latent-space policy but only uses it for one-shot imitation. And Mandlekar et al., 2019 (IRIS) which learns latent-space for RL but only offline. One can enumerate many more such examples of “X did this, but differs from this paper in this specific aspect”. But why is this new / important? What does this method do that makes it stand out / can learn tasks so difficult that no other methods can do? In my opinion, just presenting an alternative (but not too different from existing works) to a problem without clearly stating and demonstrating the “why” is not sufficient.

Relatedly, the experimental result is lacking. Just comparing with a single baseline on such a densely studied topic is clearly not enoughs. One may argue that none of the closely related works discussed makes exactly the same assumption as the presented work, but many of them can be adapted and used for a fair comparison. The experiment result is also a good place to show why the presented method matters. It’d be great to show that the method can outperform other methods on real robot platforms (e.g., table-top manipulation).

Minor:
- Figure 2 is somewhat confusing. I only understood what it means after going through section 3 a few times. The colors and arrows don’t really help explain the figure.
- What is the “no-prior” baseline?

---

> ### Author Response · Authors · 2022-11-28
> **Response to Reviewer TeiQ**
>
> We thank the reviewer for their thoughtful comments. We updated Figure 2 in the paper with the aim to improve clarity wrt hierarchical decision-making in our framework. The "no-prior" baseline is comparable to CoMic (i.e., learning a high-level policy on a task with a fixed pre-trained policy, but without using a latent space prior).
>
> Regarding further support for the importance of our proposed method, we are currently finalizing experiments on two additional domains; see our global response for further details, in which we also argue that state-of-the-art approaches developed on popular offline RL benchmarks might not be a good fit for our primary domain of interest.

---

### Review · Reviewer_E714 · 2022-11-09

**Summary Of Contributions:**

This work explores how a “latent space prior” can be learned from observation data to accelerate reinforcement learning. First, a representation of latent actions $z$ is learned with an auto-encoder on observation trajectories. Next, a sequence model is learned with an auto-regressive model (implemented with a Transformer). A separate “low-level policy” is learned to translate high level actions $z$ into low level actions.

The paper explores a few ways which such a setup can be used to accelerate policy learning on different transfer tasks:

1. First, it proposes to use the latent space prior together with the low level policy as an exploration policy.
2. Second, it proposes to use the latent space prior as regularization while learning a high level policy by biasing the learned policy towards the prior
3. Third, it proposes to use the prior as temporally extended options by learning a policy that sets the initial latent state for the prior, from which $k$ time-steps of high level actions are sampled.
4. Finally, the latent space priors can be used for planning, by sampling sequences of high level actions and decoding them into states using the auto-encoder and evaluating the rewards of various sampled sequences.

The strength of these approaches are evaluated on a simulated humanoid robot. The latent space prior is trained on motion capture data, and each of the above methods is tested across four tasks. The method works well and compares strongly to the baselines on all of the tasks.

**Audience:**

Yes

**Broader Impact Concerns:**

I do not believe there are any broader impact concerns with this paper.

**Claims And Evidence:**

Yes

**Requested Changes:**

1. [Critical] Please add some discussion about how the latent states can be used as high level actions.
2. [Critical] I would like to see a discussion about the applicability of different approaches in stochastic environments.
3. [Critical] I would like to see a discussion about the baselines, justifying the choice of baseline and the relationship between the baselines and your method.
4. I think the related works section could be improved by adding subsection or paragraph headers.

**Strengths And Weaknesses:**

## Strengths.

1. The paper is well written and quite clear. I appreciate the care that is taken to put it in context with respect to prior works.
2. Given that a latent space prior is learned, I think that the four ways of using the prior (exploration, regularization, options, and planning) make a lot of sense.
3. The performance of the method seems strong.

## Weaknesses.

1. I think that the paper is a bit unclear about whether the latent space $z$ corresponds to latent states or actions. In some parts they are referred to as states (which makes sense, since they are learned by auto-encoded states), and in others refer to them as actions.
2. The above distinction matters since state transitions in an MDP can be controlled by actions as well as by environmental stochasticity. Does this approach work in general in stochastic environments? I suspect that at least with the planning method, high level “actions” that correspond to lucky environment transitions that give high reward may be selected, leading to issues.
3. The relationship with CoMic should be made very clear, since it is the main comparison.

---

> ### Author Response · Authors · 2022-11-28
> **Response to Reviewer E714**
>
> We highly appreciate the reviewer's feedback. We updated Figure 2 in the paper and some of the text (highlighted in blue in the updated submission) to clarify the role of latent states when learning high-level policies. In summary, a high-level action refers to a near-future latent state, which in turn encodes successive demonstration states. We also clarified the relationship to CoMic at the beginning of the experimental section and added paragraph headings to the Related Work section.
>
> In stochastic environments, we expect our method to also be applicable. In contrast to decision transformers, which do indeed struggle in stochastic environments (http://arxiv.org/abs/2205.15967), our priors are not conditioned on returns. Even in our setup, our low-level policy cannot perfectly reenact every sequence sampled from the prior. In the planning experiment, we replan frequently based on the current state in a receding horizon fashion, which should help when dealing with stochastic transitions. Finally, in our top-level response we include experiments on the Frank Kitchen domain, which has deterministic underlying dynamics but features observation noise.

---

### Review · Reviewer_HF7j · 2022-11-16

**Summary Of Contributions:**

This paper presents a method to leverage demonstrations with only sequences of observations for learning a latent space prior modeled with transformers, train a low-level policy on top of the latent representation, and then performs down-stream task learning via obtaining a high-level policy guided by the low-level policy. The authors show that such a method can perform well on various down-stream locomotion tasks leveraging publicly available motion capture data for pre-training.

**Audience:**

Yes

**Claims And Evidence:**

Yes

**Requested Changes:**

1. Add more comparisons such as [1, 2].
2. Add more domains for evaluation, e.g. robotic manipulation, using human videos as the pre-training demonstrations.
3. Clarify the novelty.

**Strengths And Weaknesses:**

Strengths:

1. The idea of pre-training observation-only demonstrations is neat and the resulting three ways of leveraging the pre-trained latent priors and low-levels, i.e. exploration, regularization and options are reasonable and nicely formulated.

2. The experiment results clearly show that the method can outperform no prior and prior method CoMic low-level policy,

Weaknesses:

1. I think the comparison in the paper is not thorough. One very related direction is leveraging human videos in robot learning, which follows exactly the same motivation of the work. I think the authors should consider discussing and comparing to these works such as [1,2].

2. The domains shown in the paper are limited to simulated locomotion. I think the method should really shine in areas where actions of the demonstrations are hard to obtain, e.g. human videos as discussed above.

3. The idea of the paper is not particularly novel. The autoregressive latent priors [3], hierarchical skill discovery for down-stream task learning [4] and observation-only demonstration learning [1,2] have all been explored before. The authors seem to piece them together to tackle a less explored task (simulated locomotion), which makes the work less exciting.

[1] Chen, Annie S., Suraj Nair, and Chelsea Finn. "Learning generalizable robotic reward functions from" in-the-wild" human videos." arXiv preprint arXiv:2103.16817 (2021).

[2] Yamada, Jun, et al. "Task-Induced Representation Learning." arXiv preprint arXiv:2204.11827 (2022).

[3] Sontakke, Sumedh A., et al. "Video2Skill: Adapting Events in Demonstration Videos to Skills in an Environment using Cyclic MDP Homomorphisms." arXiv preprint arXiv:2109.03813 (2021).

[4] Ajay, Anurag, et al. "Opal: Offline primitive discovery for accelerating offline reinforcement learning." arXiv preprint arXiv:2010.13611 (2020).

---

> ### Author Response · Authors · 2022-11-28
> **Response to Reviewer HF7j**
>
> We appreciate the reviewer's feedback and suggestions. We point to our top-level comment which introduces experiments on two additional domains and hence allows for direct comparison to other approaches as suggested by the reviewer (e.g., OPAL [4]; references are numbered as provided by the reviewer).
>
> We respectfully argue that learning behavior from videos is not the subject of our study, but we'd be happy to include a short discussion on this in the conclusion. In summary, we think that video demonstrations would pose additional, albeit orthogonal, challenges in our setup. First, demonstrations could not be easily grounded in the simulator, requiring methods like [5] or inspiration from [1] for acquiring a low-level policy via imitation learning. Second, latent states might capture irrelevant information (as identified by [2], which focuses on state representation learning in multi-task settings).
>
> With respect to novelty, we note that, to our knowledge, no prior work provides a similarly flexible framework for auto-regressive priors, and additionally demonstrates it on a challenging domain with high-dimensional observation and action spaces. Video2Skill [3], mentioned by the reviewer, does not include online reinforcement learning experiments, and utilizes generative sequence models in order to map from videos to the robot's MDP. The downstream tasks in [4] are taken from popular offline RL benchmarks, i.e., demonstrations closely match the final task.
>
> [5] YuXuan Liu, Abhishek Gupta, Pieter Abbeel, Sergey Levine (2018) [Imitation from Observation: Learning to Imitate Behaviors from Raw Video via Context Translation](http://arxiv.org/abs/1707.03374) _arXiv:1707.03374 [cs]_

---

### Author Response · Authors · 2022-11-28
**Top-level Response**

We'd like to thank the reviewers for the useful feedback. Both reviewer TeiQ and HF7j suggested we include further baselines to strengthen our claims. We ran a series of experiments on the D4RL maze and Franka kitchen robot tasks. Our experiments are still ongoing, so we would like to ask the reviewers to provide us with an additional week to gather final results and include them in the paper. We'll share preliminary but encouraging results below, which we believe demonstrate the generality of our approach.

At the same time, we'd like to point out that we expect methods like SPiRL [1] or OPAL [2] that also learn low-level policies from demonstrations to not be a good fit for our setting because
- We do not have ground truth actions available in our demonstration dataset. In our submission we include experiments with MoCapAct [3], where low-level policies are trained with behavior cloning, but with the setup from their paper the results were discouraging (Fig. 15).
- Popular offline benchmarks like D4RL, for which these methods were originally developed, are set up so that the dataset covers the required downstream behavior. We hypothesize that for this reason, state-conditioned priors turn out to be very effective in the aforementioned papers.

## AntMaze
We use the same training data and task versions as in OPAL [2], and follow them in excluding the robot's X/Y location for low-level policy training. We obtained best results when training the state encoder and policy together on the offline data (i.e., in a similar fashion than NPMP [4]). The results on the "medium" maze highlight how latent space priors can boost learning quite dramatically in a dense-reward setting (3 seeds for each run; "no prior" starts reaching goals after about 3M steps (not shown)): **[Figure](https://imgur.com/92jzrGu)**.

Since our priors are state-independent, we can further easily apply the low-level policy and priors acquired on the medium dataset to the large maze (we're comparing to OPAL trained on the large maze's dataset): **[Figure](https://imgur.com/Dc8dMu3)**.

In the sparse-reward settings for the medium maze we occasionally reach goal states during training but haven't been able to reliably reach them in evaluation rollouts yet. We hypothesize that state-dependent priors as used in [1] and [2] would be particularly helpful in sparse-reward settings in order to guide the high-level policy towards relevant behavior in specific positions of the maze. We regard this as a possible extension of our work and will mention this in the conclusion.

## Franka Kitchen
We follow the setup from SPiRL [1], with the difference that our policies receive the last 4 observations as input since the environment is not markov (joint velocities are not observed). As for AntMaze, we jointly train the state encoder and low-level policy. We evaluate latent space priors both for exploration and as options, and obtained promising results with the latter (each run with 3 seeds): **[Figure](https://imgur.com/72WCpHX)**.

For reference, [1] reports a mean return of approx. 2.75 on this task (Fig. 4).


[1] Karl Pertsch, Youngwoon Lee, Joseph J. Lim (2020) [Accelerating Reinforcement Learning with Learned Skill Priors](http://arxiv.org/abs/2010.11944) _Conference on Robot Learning_

[2] Anurag Ajay, Aviral Kumar, Pulkit Agrawal, Sergey Levine, Ofir Nachum (2021) [OPAL: Offline Primitive Discovery for Accelerating Offline Reinforcement Learning](https://openreview.net/pdf?id=V69LGwJ0lIN) In _International Conference on Learning Representations_

[3] Nolan Wagener, Andrey Kolobov, Felipe Vieira Frujeri, Ricky Loynd, Ching-An Cheng, Matthew Hausknecht (2022) [MoCapAct: A Multi-Task Dataset for Simulated Humanoid Control](http://arxiv.org/abs/2208.07363) _arXiv:2208.07363_

[4] Josh Merel, Leonard Hasenclever, Alexandre Galashov, Arun Ahuja, Vu Pham, Greg Wayne, Yee Whye Teh, Nicolas Heess (2019) [Neural Probabilistic Motor Primitives for Humanoid Control](https://openreview.net/forum?id=BJl6TjRcY7) _International Conference on Learning Representations_

---

> ### Author Response · Authors · 2022-12-05
> **Paper updated with results on offline datasets**
>
> We updated the paper with the experiments on the ant maze and kitchen robot experiments discussed above (highlighted in blue). We are confident that by comparing to relevant prior work on skill learning from (offline) demonstrations this way, we demonstrate that our approach achieves competitive results and is applicable to various domains.

---

### Decision · Action_Editors · 2023-02-19

**Recommendation:** Accept as is

**Comment:**

This paper studies leveraging offline datasets for learning “latent space prior” using transformers and using that to train a low-level policy. The paper explores several methods of integrating the latent space priors for the downstream tasks, for example, by augmenting the exploration policy with sequences sampled from the prior, regularizing the policy towards distributions predicted by the prior or planning with HRL/options framework.

The paper is very well-written and easy to read. The novelty of the method is limited, but the experimental results are interesting. Nevertheless, this paper was borderline, with reviewers having confusion about the novelty and experimental protocol presented in this paper. However, the authors address most of the concerns raised by the reviewers during the rebuttal phase. The authors have addressed some of the important concerns the reviewers raised regarding baselines and experiments on other domains. They have clarified some of the concerns related to novelty as well. However, there are still outstanding comments raised by the Reviewer TeiQ about the experiments on real-world tasks. However, running experiments on real robots and including them in the paper would be beyond the scope of this rebuttal. As it stands, the experiments on the simulation environments used in the latest version of the paper should be sufficient for publication.

**Audience:**

Learning control policies from demonstrations is an important area of reinforcement learning, especially for the hard exploration problems. This is an important research topic in machine learning and I think would be of interest of the TMLR community.

**Claims And Evidence:**

The experiments and results presented in this paper are clear but only focused on limited set of simulation environments. There are no experiments on real-world tasks and all the environments used in this paper are deterministic. Overall the claims made in this paper are correct and supported by the experiments on simulated deterministic continuous control environments. The paper shows that the proposed method outperforms the baselines on three different benchmarks:
- Ant-maze
- Locomotion
- Franka Kitchen